

# 1 A new method for amino acid geochronology of the bivalve shell *Arctica*

# 2 *islandica*

Martina L. G. Conti[1], Paul G. Butler[2], David J. Reynolds[2], Tamara Trofimova[2], James D. Scourse[2],
Kirsty E. H. Penkman[1]
[1]Department of Chemistry, University of York, York, YO10 5DD, United Kingdom
[2]Centre For Geography And Environmental Science, University Of Exeter, United Kingdom
*Correspondence to*: Martina L.G. Conti (martina.conti@york.ac.uk)

## 8 Abstract

The bivalve mollusc *Arctica islandica* can live for hundreds of years, and its shell has provided a valuable resource for
sclerochronological studies and geochemical analyses for understanding palaeoenvironmental change. Shell specimens
recovered from the seabed need to be dated in order to aid sample selection, but existing methods using radiocarbon
dating or crossdating are both costly and time-consuming. We have investigated amino acid geochronology (AAG) as
a potential alternative means of providing a less costly and more efficient rangefinding method. In order to do this, we
have investigated the complex microstructure of the shells, as this may influence the application of AAG. Each of the
three microstructural layers of *A. islandica* have been isolated and their protein degradation examined (amino acid
concentration, composition, racemisation and peptide bond hydrolysis). The intra-crystalline protein fraction was
successfully extracted following oxidation treatment for 48 h, and high temperature experiments at 140°C established
coherent breakdown patterns in all three layers, but the inner portion of the outer shell layer (iOSL) was the most
appropriate component due to practicalities. Sampling of the iOSL layer in Holocene shells from early and late
ontogeny (over 100-400 years) showed that the resolution of AAG is too low in *A. islandica* for within-shell age
resolution. However, analysis of 19 subfossil samples confirmed that this approach could be used to establish a relative
geochronology for this biomineral throughout the whole of the Quaternary. In the Late Holocene the temporal
resolution is ~1500-2000 years. Relative dating of 160 dredged shells of unknown age were narrowed down using
AAG as a range finder, showing that a collection of shells from Iceland and the North Sea covered the Middle Holocene,
Late Holocene, post-medieval (1171-1713 CE) and modern day. This study confirms the value of *A. islandica* as a
reliable material for rangefinding and for dating Quaternary deposits.





**Short summary**

The mollusc *Arctica islandica* can survive for hundreds of years and its annual growth captures environmental conditions, so each shell provides a detailed climatic record. Dating is essential for sample selection, but radiocarbon and crossdating are time-consuming and costly. As an alternative, amino acid geochronology was investigated in the three aragonitic layers forming the shells. This study confirms the value of AAG as a method for rangefinder dating Quaternary *A. islandica* shells.

**1 Introduction**

*Arctica islandica* (ocean quahog) is a bivalve mollusc that inhabits the continental shelf seas across the North Atlantic region (MarLIN database). It presently lives across subpolar latitudes of the North Atlantic region of Europe from the English Channel to the White Sea, and in North America from Virginia to Nova Scotia (MarLIN database, Schöne, 2013). Its Quaternary subfossil shells are also found in ancient sediments in Northern Europe and in the Mediterranean Sea (Malatesta and Zarlenga, 1986; Eyles et al., 1994; Crippa et al., 2019). *Arctica islandica* has been routinely used for palaeoclimate and palaeoceanographic studies due its exceptionally long life (>500 years maximum longevity; Butler et al, 2013), and its capability to capture climatological changes within its periodic accretions (i.e. growth lines; Witbaard et al., 1997; Schöne et al., 2005a; Schöne, 2013; Butler et al, 2013; Reynolds et al., 2016; Estrella-Martínez et al., 2019). The study of annual and sub-annual band growth variability within the calcium carbonate shells, termed sclerochronology, provides high-resolution detailed palaeoclimatology data spanning decades to multiple centuries (Schöne et al., 2004; Schöne and Fiebig, 2008; Dunca et al., 2009; Butler et al., 2009, 2013; Wanamaker et al., 2012; Reynolds et al., 2016; Trofimova et al., 2018; Estrella-Martinez et al., 2019; Brosset et al., 2022).

Developing sclerochronological records requires visual and statistical cross-matching across numerous samples; this endeavour can be hugely time consuming and therefore needs to be targeted appropriately, especially when dead-collected samples are of unknown age. Dating of the specimens is essential to develop accurate sclerochronological records: radiocarbon dating can be a very precise technique for Late Quaternary marine shells (back to 40,000-55,000 years). However, this is not always economically viable (Hajdas et al., 2021), especially for a large number of samples, while accurate correction for the marine reservoir effect is required and the dating uncertainty can be a few hundreds of years (Alves et al., 2018). One possible alternative is amino acid geochronology (AAG), a relative-age technique that is comparatively fast and inexpensive. AAG is applicable to mollusc shell deposits spanning the Quaternary period (e.g. Sejrup and Haugen, 1994; Davies et al., 2009; Ortiz et al., 2009; 2015; Penkman, 2010; Demarchi et al., 2013a-b, Bridgland et al., 2013), and can have high precision and resolution in tropical corals (Hendy et al., 2012).





AAG dating of biominerals is based on the natural degradation of proteins to determine age; the main processes are
racemisation (and epimerisation, both leading to an increase in amino acid D/L), peptide bond hydrolysis, and amino
acid decomposition (Hare and Mitterer, 1969). When organisms die, or when there is no more tissue turnover, these
degradation reactions occur in tandem. The inter-crystalline fraction of biominerals (Gries et al., 2009), the protein
which forms a matrix between the crystallites, is potentially more susceptible to external contamination or leaching,
and can compromise the reliability of AAG in some biominerals (Sykes et al., 1995, Penkman et al., 2008; Ortiz et al.,
2015). In some materials, a small fraction of the protein is contained within the interstitial voids of the crystal structure
and can be isolated with an oxidising pre-treatment; this is defined as the intra-crystalline (IcP) fraction (Towe and
Thompson, 1972; Sykes et al., 1995, Penkman et al., 2008; Gries et al., 2009). The IcP can be isolated with oxidation
using NaOCl (or $H_2O_2$ in some cases) and its stability against external contamination and leaching means that, in some
biominerals, it effectively operates as a closed-system. The isolation of closed-system IcP has provided reliable
chronological information in some gastropods (Penkman et al., 2008; Ortiz et al., 2015; Demarchi et al., 2013a-b;
Bridgland et al., 2013), ostracods (Ortiz et al., 2013), corals (Hendy et al., 2012; Tomiak et al., 2013; 2016), eggshell
(Brooks et al, 1990; Crisp et al., 2013), enamel (Dickinson et al., 2019; Baleka et al., 2021) and foraminifera (Wheeler
et al., 2021), but not in all biominerals (e.g. Orem and Kaufman, 2011; Torres et al., 2013; Demarchi et al., 2015).
A further complication for AAG of bivalve shells is that the different microstructural layers in bivalves are likely to be
composed of different proteins, and therefore may degrade differently. The *A. islandica* shell comprises a periostracum
and three aragonitic layers of differing crystal microstructure (Schöne, 2013): a homogeneous granular structure in the
outer portion of the outer shell layer (oOSL), a cross-acicular structure for the inner portion of the outer shell layer
(iOSL), and a cross-lamellar to cross-acicular structure for the inner shell layer (ISL; Fig. 1; Dunca et al., 2009; Schöne,
2013; Milano et al., 2017b).

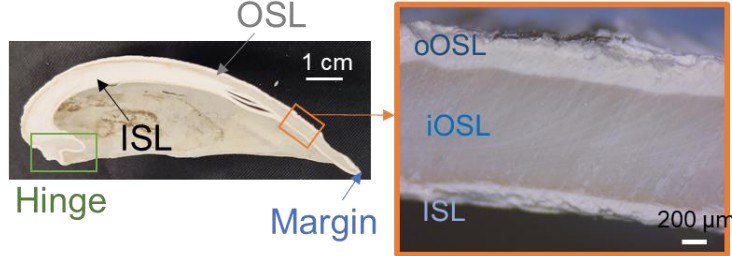


Figure 1. (Left) cross section of *Arctica islandica* showing the (right) inner shell layer (ISL), inner portion of the outer
shell layer (iOSL), and outer portion of the outer shell layer (oOSL).



*Arctica islandica, Glycymeris glycymeris, Callista chione* and *Entemnotrochus adansonianus* have shown distinct
racemisation and epimerisation rates which depend on the microstructural layer analysed (Haugen and Sejrup, 1990,
1992; Sejrup and Haugen, 1994; Goodfriend et al., 1995, 1997; Torres et al., 2013; Demarchi et al., 2015). Early
studies without chemical oxidation on *A. islandica* (i.e. combining both the inter-and any intra-crystalline fraction)
showed different epimerisation rates, AA concentrations and composition between the inner and outer layers (Haugen
and Sejrup, 1990, 1992; Sejrup and Haugen, 1994). Intra-shell variability was also high, hypothesised to be due to
microorganism attack of the protein during early stages of diagenesis, external contamination and/or leaching through
micropores (Sejrup and Haugen, 1994; Kosnik and Kaufman, 2008). A study on the use of D/L for ontogenetic studies
of unbleached shells revealed Asp D/L values higher in the umbo growth lines (laid down when the shells are young)
compared to the rim growth lines (deposited when the shell is old; Goodfriend and Weidman, 2001). A difference in
AA composition between early and late ontogeny was also observed, indicating the need of sampling standardisation;
the recommendation was to sample shells from band year 20 in the outer shell layer (Goodfriend and Weidman, 2001).
Asx D/L values (Asx indicating aspartic acid and asparagine, that cannot be distinguished due to irreversible
deamination) in unbleached *A. islandica* shells collected between 1982 and 1994 were shown to increase with age over
a 154-year chronology, highlighting that AAG can potentially help in dating sclerochronologies (Marchitto et al.,

95 2000).

Given the variability observed in unbleached *A. islandica* shell AA data, a way forward is to test for the presence of
any IcP in *A. islandica* shells, and whether it forms a closed system for individual microstructures (e.g. Torres et al.,
2013; Demarchi et al., 2013a-b, 2015; Baldreki et al., 2024). The use of IcP in AAG has not been fully investigated
on *A. islandica* (Sykes et al., 1995), and there is variety in sampling strategy for specific microstructural layers (Haugen
and Sejrup, 1990, 1992; Sejrup and Haugen, 1994; Marchitto et al., 2000; Goodfriend and Weidman, 2001). If it is
possible to isolate an intra-crystalline fraction that exhibits closed-system behaviour from any of the layers in *A.*
*islandica* shells, an IcP approach may be able to reduce the intra-shell viability, and increase accuracy.
**1.1 Aims**
We present here a new method for the preparation of aragonitic *A. islandica* shells for AAG. To develop this
methodology, the following experiments were conducted:
- optimisation of the sampling method and isolation of the three microstructural layers (Sec. 2.2);
- assessment of aragonitic mineral diagenesis via X-ray diffraction (XRD) analysis (Sec. 3.1);
- testing for the existence of an intra-crystalline protein fraction via oxidation experiments (Sec. 3.2);



-    testing for closed-system behaviour of *A. islandica* through controlled high temperature decomposition
110         experiments and assessment of the amino acid degradation patterns (Sec. 3.3);

-    assessment of any change in amino acid composition and D/L values with ontogeny (Sec. 3.4);
-    optimised method and recommendations for IcPD analysis of *A. islandica* (Sec. 3.5);
-    analysis of multiple independently-dated subfossil shells to develop an initial AAG framework for *A. islandica*
114         in the North Atlantic Ocean (Sec. 3.6);

-    age rangefinding of undated shells collected during research cruise DY150 of RRS *Discovery* in spring 2022
116         (Sec. 3.7).

**2 Materials and methods**
**2.1 *A. islandica* specimens**
In total, 19 *A. islandica* subfossil samples from the North Sea and Iceland were obtained for the bleaching and high
temperature experiments, ontogenetic trends and initial framework; these spanned in age from modern to ~2.1-2.2 Ma
and were independently dated with radiocarbon ($^{14}$C), AAG on other biominerals (see Table 1 for details; Fig. 2), and
sclerochronological crossdating (S; Table 1). In addition, 160 *A. islandica* shells of unknown age, incorporating
samples from both the North Sea and the North Icelandic shelf, were analysed for rangefinding (Sec. 3.7; Fig. 2).



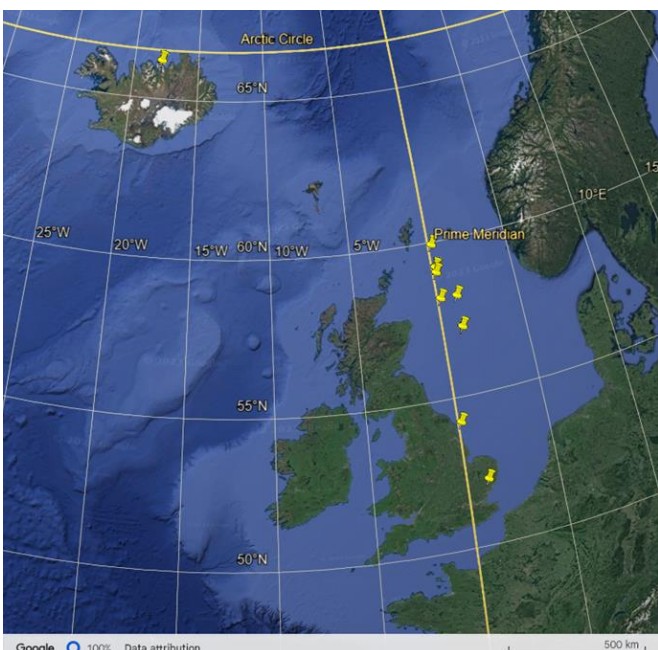


Figure 2. Location of the *A. islandica* samples analysed in this work. Map created using © Google Earth.

Table 1. Overview of the *A. islandica* shells analysed in this study. Methods of dating: AAG: amino acid geochronology; [14]C: radiocarbon; S: sclerochronologically crossdated. * Note: beach-collected samples could range thousands of years in age (e.g. Dominguez et al., 2016). § Further detail on radiocarbon dates are in Supplementary information Table S1. ※ Further information about sampling location are in Supplementary information Table S2.

| Sample name | Number of shells and code | Locality | Latitude | Longitude | Estimated age | Independent previous dating | Reference for previous dating | Experiment |
|---|---|---|---|---|---|---|---|---|
| ArBrMod | n=1 N/A | Bridlington beach, UK | 54° 4' N | 0° 11' W | Modern?*, beach collected July 2021 | N/A | N/A | pXRD (Sec. 3.1), framework (Sec. 3.6), rangefinding (Sec. 3.7) |
| ArPe, ArPe2 | n=2 N/A | North Sea off Peterhead | 58° 37' N | 1° 27' E | Modern, live-collected, trawled at -114m depth in 2018 | N/A | schnecken-und-muscheln.de | Bleaching (Sec. 3.2), high temperature (Sec. 3.3), framework (Sec. 3.6), |



| | | | | | | | | rangefinding (Sec. 3.7) |
|---|---|---|---|---|---|---|---|---|
| ArNsM1 | n=1 0401381R | North Sea | 59° 23.10' N | 0° 31.00' E | 1865-2004 CE | S | Butler et al., 2009 | Ontogenetic trends (Sec. 3.4), framework (Sec. 3.6), rangefinding (Sec. 3.7) |
| ArNsM2 | n=1 0401422L | North Sea | 59° 23.10' N | 0° 31.00' E | 1874-2004 CE | S | Butler et al., 2009 | Ontogenetic trends (Sec. 3.4), framework (Sec. 3.6), rangefinding (Sec. 3.7) |
| ArNsM3 | n=1 0401423L | North Sea | 59°23.10' N | 0°31.00' E | 1908-2004 CE | S | Butler et al., 2009 | Ontogenetic trends (Sec. 3.4), framework (Sec. 3.6), rangefinding (Sec. 3.7) |
| ArNs0246 | n=1 0400246 | North Sea | 59° 7.5' N | 0° 10.0' E | 1867-2004 CE | S | Butler et al., 2009 | pXRD (Sec. 3.1), ontogenetic trends (Sec. 3.4), framework (Sec. 3.6), rangefinding (Sec. 3.7) |
| ArIcP2 | n=1 061683M | Iceland | 66° 31.59' N | 18° 11.74' W | 1397-1713 CE | S | Butler et al., 2013 | Ontogenetic trends (Sec. 3.4), framework (Sec. 3.6), rangefinding (Sec. 3.7) |



| | | | | | | | | |
|---|---|---|---|---|---|---|---|---|
| ArIcP1 | n=1 061682M | Iceland | 66° 31.59' N | 18° 11.74' W | 1171-1391 CE | S | Butler et al., 2013 | pXRD (Sec. 3.1), ontogenetic trends (Sec. 3.4), framework (Sec. 3.6), rangefinding (Sec. 3.7) |
| ArIc617 | n=1 061617 | Iceland | 66° 31.59' N | 18° 11.74' W | 2841±33 $^{14}$C yr BP 2545-2119 cal yr BP 2σ | $^{14}$C § | n/a § | Framework (Sec. 3.6), rangefinding (Sec. 3.7) |
| ArIc711 | n=1 061711 | Iceland | 66° 31.59' N | 18° 11.74' W | 2938±33 $^{14}$C yr BP 2678-2292 cal yr BP 2σ | $^{14}$C § | n/a § | Framework (Sec. 3.6), rangefinding (Sec. 3.7) |
| ArIc407 | n=1 061407 | Iceland | 66° 31.59' N | 18° 11.74' W | 3411±37 $^{14}$C yr BP 3223-2817 cal yr BP 2σ | $^{14}$C § | n/a § | Framework (Sec. 3.6), rangefinding (Sec. 3.7) |
| ArIc746 | n=1 061746 | Iceland | 66° 31.59' N | 18° 11.74' W | 3535±36 $^{14}$C yr BP 3364-2975 cal yr BP 2σ | $^{14}$C § | n/a § | Framework (Sec. 3.6), rangefinding (Sec. 3.7) |
| ArIc305 | n=1 061305 | Iceland | 66° 31.59' N | 18° 11.74' W | 4222±40 $^{14}$C yr BP 4257-3826 cal yr BP 2σ | $^{14}$C § | n/a § | Framework (Sec. 3.6), rangefinding (Sec. 3.7) |
| ArNs0658 | n=1 010658 | Fladen Ground (North Sea) | 58° 50' N | 0° 21.35' W | 7810±25 $^{14}$C yr BP 8340-8100 cal yr BP 2σ | $^{14}$C (Marine13 calibration) | Estrella-Martinez, 2019 | Framework (Sec. 3.6), rangefinding (Sec. 3.7) |
| ArNsP1 | n=1 10660 | Fladen Ground (North Sea) | 58.831° N | −0.356° E | 7801±29 $^{14}$C yr BP 8330-8070 cal yr BP 2σ | $^{14}$C (Marine13 calibration) | Estrella-Martinez, 2019 | Ontogenetic trends (Sec. 3.4), framework (Sec. 3.6), |



| | | | | | | | | |
|---|---|---|---|---|---|---|---|---|
| | | | | | | | | rangefinding (Sec. 3.7) |
| ArNsP2 | n=1 10682 | Fladen Ground (North Sea) | 58.831° N | −0.356° E | 7794±24 $^{14}$C yr BP 8320-8060 cal yr BP 2σ | $^{14}$C (Marine13 calibration) | Estrella-Martinez, 2019 | Ontogenetic trends (Sec. 3.4), framework (Sec. 3.6), rangefinding (Sec. 3.7) |
| ArNsP3, ArNs0684 | n=1 10684 | Fladen Ground (North Sea) | 58.831° N | −0.356° E | 7752±23 $^{14}$C yr BP 8280-8020 cal yr BP 2σ | $^{14}$C (Marine13 calibration) | Estrella-Martinez, 2019 | pXRD (Sec. 3.1), ontogenetic trends (Sec. 3.4), framework (Sec. 3.6), rangefinding (Sec. 3.7) |
| ArWey | n=1 N/A | Weybourne Crag, UK | 52° 56.55' N | 1° 08.33' E | Early Pleistocene (2.2-2.1 Ma) | AAG on *Bithynia* opercula and *Nucella*, biostratigraphic evidence | Preece et al., 2020 | pXRD (Sec. 3.1), bleaching (Sec. 3.2), framework (Sec. 3.6) |
| Multiple names, see Table S2 | n=73 | Fladen Ground (North Sea) | Various ※ | Various ※ | Unknown | None | N/A | Rangefinding (Sec. 3.7) |
| Multiple names, see Table S2 | n=87 | North Icelandic shelf | Various ※ | Various ※ | Unknown | None | N/A | Rangefinding (Sec. 3.7) |




**2.2 Sampling**


Each individual shell was sectioned from the umbo to the margin with an IsoMet 1000 precision cutter. After slicing, the
shells were sonicated in deionised water (18.2 MΩ cm⁻¹) until no residue was observed (3 min, 2-3 washes). After air drying,
the periostracum (if present), was removed by drilling with an abrasive rotary burr on a hand-held rotary tool (Dremel drill).
Each layer (oOSL, iOSL and ISL; Fig. 1), was sampled by drilling using a Dremel drill equipped with a stainless-steel diamond-
coated drill bit with a small sphere or cylindrical tip. Following the experiments in sections 3.3 and 3.4, the iOSL layer was
chosen for building the AAG framework. To check changes in amino acids with ontogeny (e.g. with the biological age of the
shell; Sec. 3.4), the iOSL from early and late ontogeny within one shell (Table 1) was sampled: the former near the hinge and
the latter close to the ventral margin of the shell, likely containing a few annual growth increments. To build the AAG
framework (Sec. 3.7), intact Quaternary shells were subsampled near the margin of the shell where the iOSL was thickest.
Fragmented shells were sampled where the iOSL was thickest, for ease of sampling. Between each sample the drill tip was
cleaned in a 0.6 M HCl (Fisher, analytical grade) solution and MeOH (Fisher, HPLC grade) to reduce cross-contamination of
samples.

**2.3 Bleaching procedure**


Following protocols developed by Penkman et al., (2008), approximately 20-30 mg of powder was transferred to a 2 mL plastic
microcentrifuge tube (Eppendorf) and NaOCl (12 %, VWR, 50 uL mg⁻¹ of sample) was added. Samples were oxidised for 24-
72 h for bleaching experiments (Sec. 3.3). Following the results of these experiments, the iOSL layer of all other subfossil
samples was oxidised for 48 h. After the allotted time, the NaOCl was removed, and the powder was washed six times with
deionised water (18.2 MΩ cm⁻¹) and once with MeOH (Fisher, HPLC grade). The samples were left to air dry for one to two
days.

**2.4 High temperature experiments**


High temperature experiments were carried out in a BinderTM ED23 oven set to 140 °C. To the bleached powder (10-20 mg),
300 μL of deionised water (18.2 MΩ cm-1) was added in a glass vial (Penkman et al., 2008). The samples were exposed to
high temperature conditions of 140 °C for 8, 24 and 48 h. After this time, the water was carefully removed and the powder
was left to air dry for 1-2 days.

**2.5 Isolation of free (FAA) and total hydrolysable amino acids (THAA)**


Following bleaching and in some cases high temperature exposure, the dry powder (1-10 mg) was split between free amino
acids (FAA) and total hydrolysable amino acids (THAA; Penkman et al., 2008). The FAA were demineralised in 2 M HCl





(10 uL mg$^{-1}$ of sample, minimum possible volume) and dried over a rotary vacuum concentration (Christ RVC 2-25 CDplus,
1300 rpm). The THAA samples were hydrolyzed in 7 M HCl (20 uL mg-1 of sample) and heated at 110 °C for 24 h to
hydrolyse the peptide bonds. The samples were then dried in a rotary vacuum concentrator.

**2.6 UHPLC-FLD analysis**

Samples were rehydrated with a solution containing an internal standard - L-homo-arginine (0.01 mM), sodium azide (1.5
mM) and HCl (0.01 M) - to enable quantification of the amino acids. Analysis of chiral amino acid pairs was achieved using
an Agilent 1200 Series HPLC fitted with an Agilent Eclipse Plus C$_{18}$ column (4.6 x 100 mm, 1.8 um particle size) and
fluorescence detector (excitation wavelength = 230 nm, emission wavelength = 445 nm), using a UHPLC method modified
from Crisp (2013; Table 2). The binary mobile phase consisted of: (A) sodium acetate buffer (23 mM sodium acetate
trihydrate, sodium azide, 1.3 μM EDTA, adjusted to pH 6.00 ± 0.01 with 10% acetic acid and sodium hydroxide), and (B)
92.5:7.5 methanol:acetonitrile. Table 2 reports the mobile phase, flow rate and temperature gradient of the separation. Data
processing was performed on ChemStation and data analysis on Excel; all data discussed in this paper is reported in
Supplementary information, Table S2. The Crisp (2013) method, is able to separate the L and D enantiomers of 14 amino
acids.
Table 2. Gradient of mobile phase, flow rate and column temperature for the UHPLC-FLD method. ‡ indicates that the
parameter does not change at the referred timepoint.

| Time / min | % A / sodium acetate buffer | % B / 92.5:7.5 methanol:acetonitrile | Flow rate / mL min$^{-1}$ | Column temperature / °C |
|---|---|---|---|---|
| 0.0 | 90 | 10.0 | 1.25 | 25 |
| 8.8 | 82.0 | 18.0 | 1.25 | ‡ |
| 10 | 82.0 | 18.0 | 1.25 | 28 |
| 11 | 82.0 | 18.0 | 1.25 | 28 |
| 23 | 78.3 | 21.7 | 1.25 | 28 |
| 25 | 78.3 | 21.7 | 1.25 | 28 |
| 32 | 75.2 | 24.8 | 1.25 | 28 |
| 34.5 | 74.0 | 26.0 | 1.25 | 28 |
| 36 | 65.0 | 35.0 | 1.25 | 28 |
| 50 | ‡ | ‡ | 1.30 | 25 |
| 56 | 50.0 | 50.0 | 1.30 | 25 |
| 62 | 2.0 | 98.0 | 1.30 | 25 |
| 67 | 95.0 | 5.0 | 1.25 | 25 |



**2.7 Powder XRD analysis**

Powder X-ray diffraction analysis was carried out on a selection of samples (Table 1) using a Bruker Panalytical Aeris Powder XRD, scanned between 0-70° 2θ using a 0.2-degree increment per second. The scan axis was Gonio, source filter was Beta nickel, beam mask was set to 20, beam knife to high, and antiscatter was 9 mm. The samples analysed were powdered either by a rotary burr on a drill (section 2.2) or by homogenising to a fine powder with an agate pestle and mortar. Homogenised *Cepea* spp. shells were used as aragonite standards and modern ostrich eggshell (OES) as calcite standard.

**3 Results and discussion**

The multilayer nature of *A. islandica* (comprising the oOSL, iOSL and ISL; Fig. 1) means that there are likely to be protein differences between layers. This will dictate the original amino acid concentration and composition, and therefore their diagenesis, with impacts on D/L values and AAG. Initially we present an assessment of the mineralogy (Sec. 3.1), followed by  the results from bleaching (Sec. 3.2) and heating experiments on the three microstructural layers (Sec. 3.3), assessing the amino acid composition, concentration and D/L values. Ontogenetic trends on modern and subfossil shells are presented in section 3.4. Recommendations for the method for AAG analysis of *A. islandica* (Sec. 3.5) are followed by an initial AAG framework over the Quaternary period (Sec. 3.6), and application of the method to age rangefinding undated shells (Sec. 3.7).

**3.1 Assessment of mineral diagenesis**

Aragonite, the polymorph of $CaCO_3$ that makes up the shells of *A. islandica*, can convert into calcite over geological timescales or under stress (Brand and Morrison, 1987). The transition state in the transformation of labile aragonite into calcite can have implications for the integrity of any closed system and the IcP (Preece and Penkman, 2005; Penkman, 2010). Thus, investigating the mineral composition of samples may help to identify compromised samples; this can be done using X-ray diffraction. In order to understand any potential changes to the $CaCO_3$ structure, powder XRD was carried out on a selection of samples of a variety of ages to qualitatively assess the presence of aragonite and/or calcite (Table 1).

The diffractograms of modern (ArNs0246), medieval, Mid-Holocene (Walker et al., 2019) and Early Pleistocene shells that were drilled show a small peak of calcite (theta 29°) in the mainly aragonitic shells (Fig. 3a). There is no clear pattern between the age of the sample and the presence or absence of calcite; the Early Pleistocene shell (ArWey, 2.2-2.1 Ma) shows only a very small calcite peak. It is possible that the abrasion and temperature created during the drilling process may affect the aragonitic crystal structure (Bäldreki et al., 2024). To test this, drilled powders were compared with shell chips from the same samples homogenised with a pestle and mortar (Fig. 3b). The chips do not show a calcite peak at theta 29° (Fig. 3b); these experiments indicate that the drilling process may cause some transformation of aragonite into calcite. However the drilling



203  process is necessary in order to remove the periostracum and isolate and sample the required layers for AAG. Therefore, it is

204  important to use the lowest speed possible and avoid applying extreme pressure when sampling using a rotary drill.

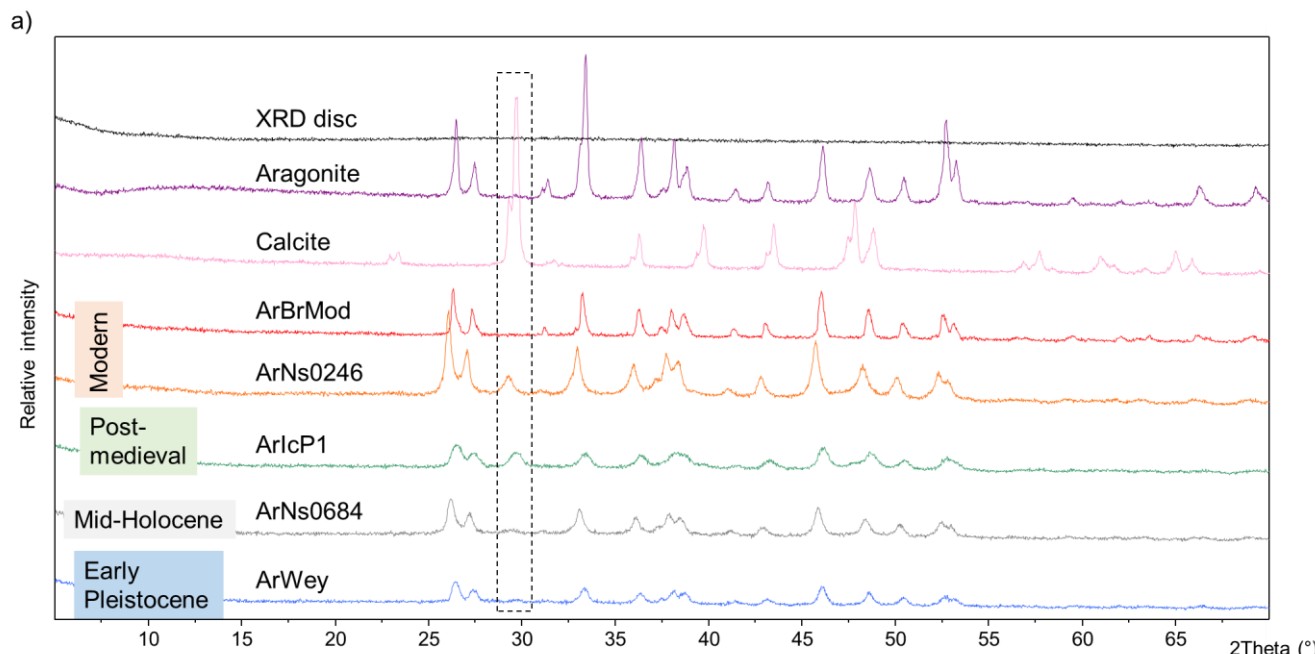

205





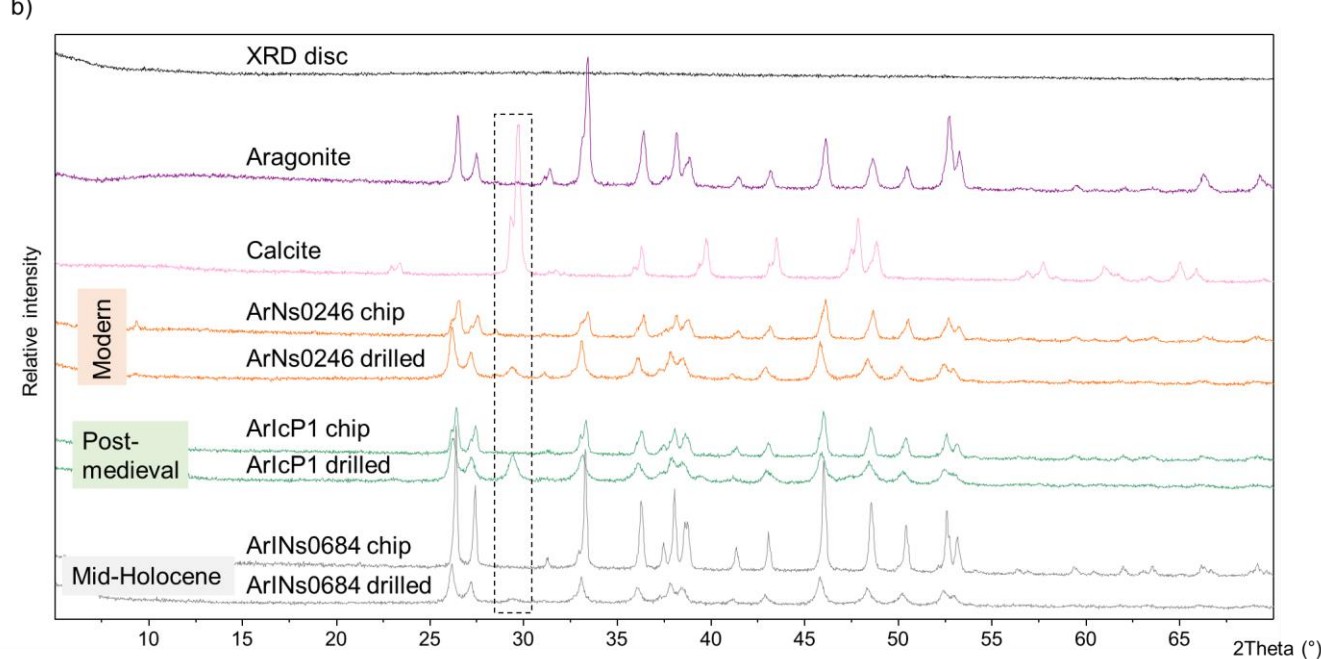

Figure 3. (a) pXRD spectra of *A. islandica* shells of various ages, powdered using a drill; (b) pXRD spectra of *A. islandica* shells: in each case the same shell was drilled with a rotary burr ("drilled"), and homogenised with pestle and mortar ("chip"). The dashed area in black represents the main peak of calcite at 2θ 29°. As the drilling process may convert aragonite into calcite, it must be undertaken with care.

## 3.2 The impact of bleaching on amino acids

To test for any presence of an intra-crystalline protein fraction, bleaching experiments on each of the layers of *A. islandica* in two shells (Table 1) was undertaken: a modern sample (ArPe) and an Early Pleistocene sample (ArWey).

In the modern sample, the concentration of FAA and THAA in all layers decreases with bleaching (Fig. 4), meaning that an inter-crystalline fraction is removed. There is an initial decrease and a subsequent very small increase in concentration after 24 h in the oOSL layer, possibly indicating that the prolonged bleaching process is breaking some of the peptide bonds, increasing the concentration of amino acids from the intra-crystalline fraction. In the iOSL and ISL layers, the concentration reaches a plateau between 48 and 72 h, indicating that an intra-crystalline fraction is more resistant to bleaching than the unbleached shells and therefore requires long oxidation exposure. This isolated intra-crystalline fraction represents $17 \pm 3\%$ (all errors represent $1\sigma$ around the mean) of the oOSL, $16 \pm 5\%$ of the iOSL and $15 \pm 2\%$ of the ISLin the unbleached FAA fraction. The total concentration in the FAA fraction is two orders of magnitude smaller than in the THAA fraction; as the



sample is young there would have been little natural breakage of the peptide bonds to form free amino acids. Following an
initial drop in concentration from 0-24 h, the concentration is stable in the THAA fraction with increasing bleaching time in
all layers. This intra-crystalline fraction represents $7 \pm 0.1\%$ of the oOSL and the iOSL ($\sigma=1$), and $18 \pm 1\%$ of the ISL in the
total unbleached THAA fraction.  In summary, the FAA and THAA in the intra-crystalline fraction are stable and isolated
between 24-48 h in the modern *A. islandica* shell.
The geological formation of free amino acids through peptide bond hydrolysis is evident in the Early Pleistocene shell from
Weybourne Crag (Fig. 5). Similarly to the modern sample, the Early Pleistocene FAA and THAA decrease in concentration
with bleaching; the variability between replicates is larger so identification of the plateau is more challenging, potentially lying
between 48 h and 72 h in both the THAA and FAA fractions.  Sykes et al. (1995) noted that solid slices of modern *A. islandica*
were less susceptible to 10% NaOCl oxidation compared to the shell powder - in the latter aspartic acid concentration reached
a plateau after 10 h - indicating that the intra-crystalline fraction for the powdered shell was isolated after just 10 h of oxidation.
In our study, when the individual shell layers are isolated and powdered, a concentration plateau is only achieved between 24
and 48 h (Fig. 5); in contrast to the results from Sykes et al. (1995), we therefore suggest that bleaching for 48 hours is necessary
to securely isolate the intra-crystalline protein fraction.  In the Early Pleistocene shell, the percentage of intra-crystalline
fraction compared to the unbleached FAA fraction is $61 \pm 5\%$ in the oOSL, $38 \pm 5\%$ in the iOSL, and $70 \pm 7\%$ in the ISL; for
the THAA fraction the intra-crystalline fraction represents $49 \pm 5\%$ of the oOSL, $30 \pm 1\%$ of the iOSL, and $55 \pm 6\%$ of the
ISL.  Due to the age of the shell, it is not surprising that the majority of the FAA are intra-crystalline, with the more labile
inter-crystalline free amino acids likely to have leached out of the system (Sykes et al., 1995).  Despite the age of the shell
(~2.2-2.1 Ma), there are both inter- and intra-crystalline amino acids present (Demarchi, 2009; Penkman et al., 2011; Demarchi
et al., 2013a-b; Ortiz et al., 2015, 2018).  It is interesting to note that the ISL and oOSL layers contain a higher relative
percentage of intra-crystalline protein (oOSL = $49 \pm 5\%$, ISL = $55 \pm 6\%$), whereas the iOSL has a much lower proportion of
IcP ($30 \pm 1\%$).



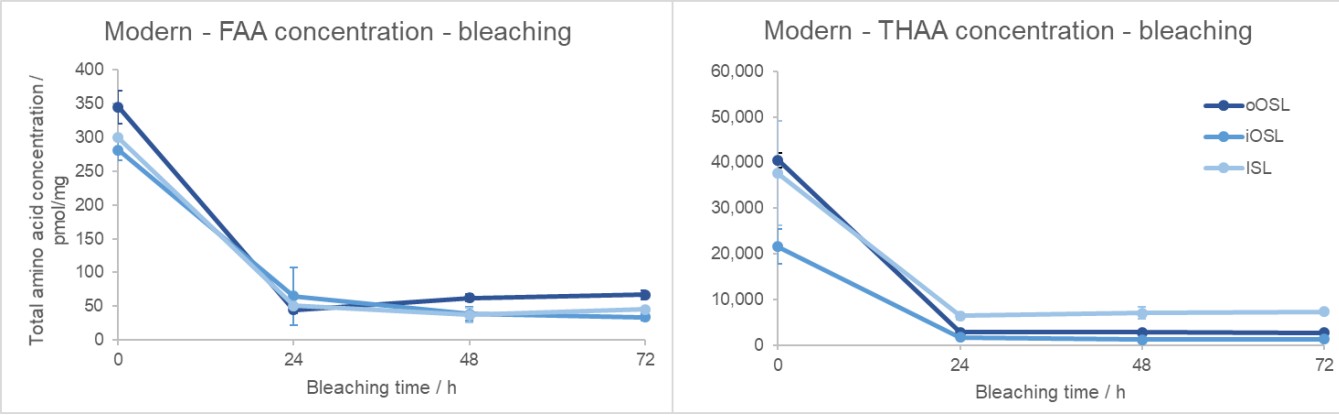

Figure 4. Decrease in total amino acid concentration upon bleaching modern *Arctica islandica* (from the North Sea off Peterhead, UK) for the oOSL, iOSL and ISL microstructural layers. Error bars indicate one standard deviation about the mean based on three replicates. Note the large drop in concentration with bleaching, with a plateau reached by 48 h.

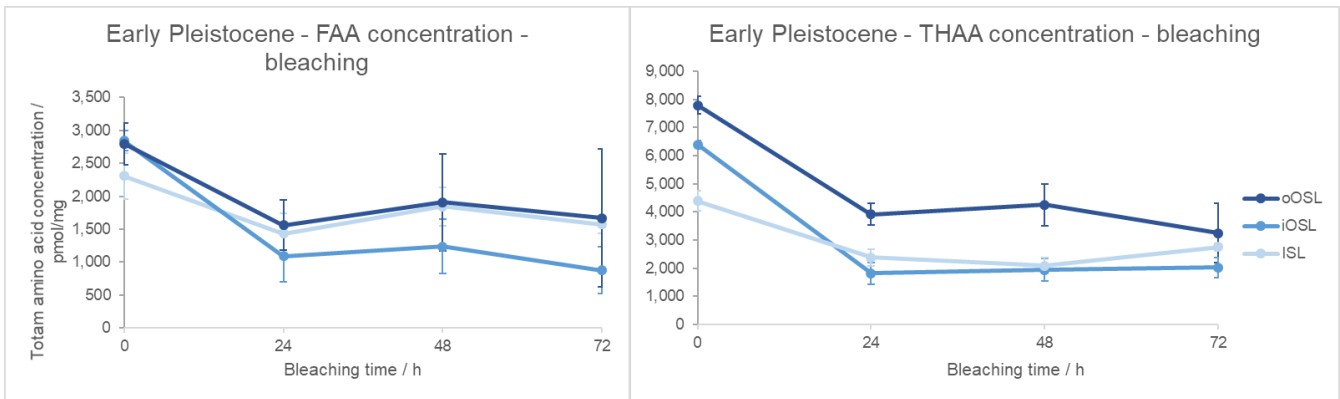

Figure 5. Change in total amino acid concentration upon bleaching Early Pleistocene *A. islandica* (from Weybourne Crag, UK) for the oOSL, iOSL and ISL microstructural layers. Error bars indicate one standard deviation about the mean based on two replicates. Note the drop in concentration with bleaching with a plateau reached by 48 h.

As the oxidation step has been shown to induce some racemisation in other mollusc shells, especially with long exposure (Penkman et al., 2008), when choosing the optimal bleaching time both concentration and racemisation have to be considered. In the modern and Early Pleistocene samples there is an initial increase in D/L for Asx (aspartic acid), Glx (glutamic acid), Ser (serine), Ala (alanine) and Val (valine) between 0-24 h bleaching, indicating that the removal of the inter-crystalline protein leaves more racemised amino acids in the IcP (Penkman et al., 2008). The D/L values reach a plateau between 24 h and 48 h, and at the 72 h timepoint the D/L values slightly increase for most amino acids, suggesting that some oxidation-induced

racemisation is taking place (Fig. 6). Nevertheless, the concentration plateau reached at 48 h in both the modern and Early

Pleistocene samples and the small change in D/L values indicates that the intra-crystalline protein fraction is effectively stable

and relatively protected from oxidation.

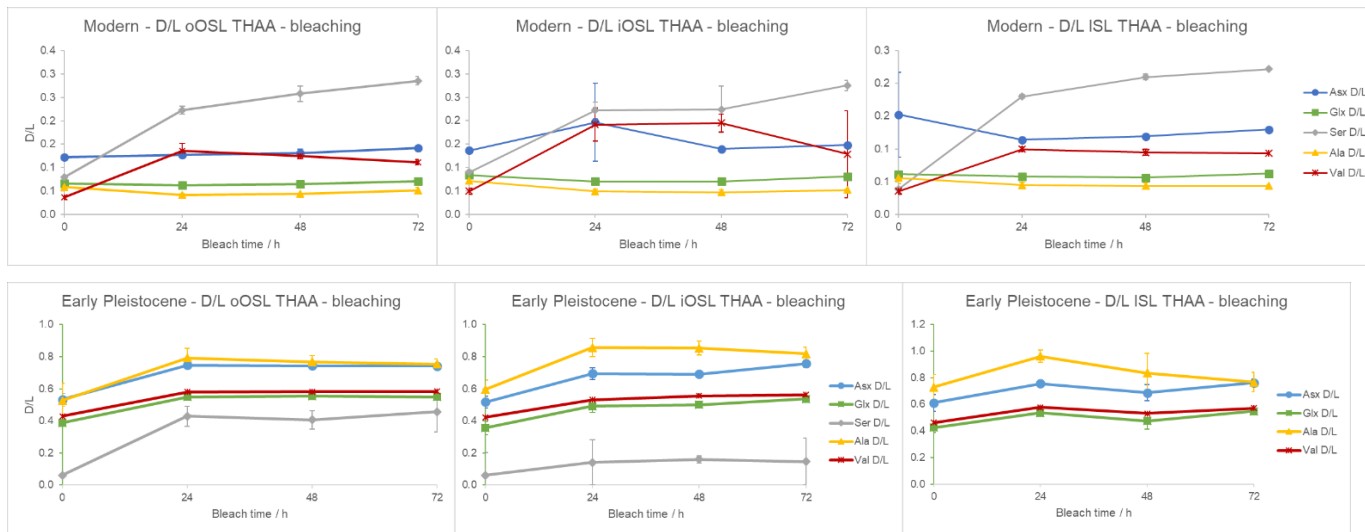

Figure 6. Mean THAA D/L of Asx, Glx, Ser, Ala & Val in *A. islandica* upon bleaching for the oOSL, iOSL and ISL

microstructural layers. Top: modern shell from Peterhead; error bars indicate one standard deviation based on three replicates.

Bottom: Early Pleistocene shell from Weybourne Crag; error bars indicate one standard deviation based on two replicates.

There is an initial increase in D/L with bleaching, but stable D/L with prolonged bleaching.

The percentage composition of each amino acid in the bleached and unbleached samples can provide information about the

nature of protein in the two fractions, if different. The composition of the unbleached shell and IcP is similar for the THAA

fraction, but some differences are present in the FAA fraction in the modern sample (Fig. 7, Supplementary information Fig.

S1). In this fraction, in the bleached samples Gly (glycine) is much higher and Ser, Ala and Arg (arginine) are lower; however

this may be due to the very low concentrations of minor amino acids, sometimes below the limit of detection. In the Early

Pleistocene sample, the composition is very similar between the unbleached and bleached fractions in both the FAA and THAA

(Supplementary information Fig. S2), confirming that the majority of amino acids in the unbleached samples are intra-

crystalline, and that much of the inter-crystalline protein fraction has leached out with time.

In addition to differences in amino acid concentration between the three layers of *A. islandica*, there are also slight differences

in composition between layers. In the THAA fraction of the modern shell bleached for 48 h, all three layers have similar

composition, with some exceptions: Ala is higher in the iOSL and Arg is lower in the ISL than in the other two layers. The

percentage composition of Gly is slightly higher in the oOSL than the iOSL and ISL, although the between-sample variability





is larger than for the other two layers which may confound the results (Fig. 7). In the Early Pleistocene shell, the FAA fraction
shows very similar composition between the three layers. In the THAA fraction, the ISL and iOSL contain higher amounts of
Asx and Glx, while Thr (threonine) is more abundant in the two OSL layers and Gly is highest in the ISL layer. There is
mostly agreement in the composition of the two shells. The higher percentage composition of Gly in the Early Pleistocene
shell is likely to be due to the natural diagenesis of Val (valine), Ser, Thr and Tyr (tyrosine) to form Gly (Vallentyne, 1964).
Ser is much lower and Ala higher in the Early Pleistocene samples: Ser is thermally unstable and can degrade to form Ala and
Gly (Vallentyne, 1964; Bada et al., 1978), while Ala can be a product of dehydration of Ser and Asx (Walton, 1998). The
overall differences in amino acid composition in both modern and Early Pleistocene shells for the three layers shows that
originally there are different proteins in the layers, which then break down at different rates; therefore, it is important to
consistently sample one layer for reliable AAG.
Haugen and Sejrup (1990) presented the percentage composition of 30 modern unbleached specimens of *A. islandica* for both
the inner and outer shell layers; as there was no separation into oOSL and iOSL, their results for the 'outer' layer are compared
to our oOSL and iOSL results (Supplementary information Fig. S1b). Additionally, Haugen and Sejrup (1990) analysed their
amino acid with ion-exchange chromatography rather than HPLC, and they do not report His (histidine), Arg and Met
(methionine). The percentage composition of the FAA and THAA fractions from the modern shell analysed here and the 30
shells from Haugen and Sejrup (1990) are very similar, with only small variations (Supplementary information Fig. S1b). In
the FAA fraction there is a lower contribution from Tyr in our data (3-4%) compared to 13-15% in the Haugen and Sejrup
(1990) shells, while Gly is higher in our data for the bleached shells (44-67%), but more comparable in the unbleached samples
(our work = 31-34%, Haugen and Sejrup, 1990 = 21-24%). In the THAA fraction the percentage composition from Haugen
and Sejrup (1990) are within error with our bleached data, while our unbleached shell has higher Gly and lower Asx and Glx.
There is remarkable similarity in percentage composition between Haugen and Sejrup (1990) results without bleaching and
our modern bleached shells; this may ultimately enable the comparison between data from samples analysed prior to and after
establishing the bleaching step in the AAG method.
A recent study compared the percentage composition of amino acids in untreated and oxidised modern shells, including 12%
NaOCl treatment on powdered shell where the layers had been homogenised (Huang et al., 2023). Similar to our results, Gly,
Asx and Glx were the dominant amino acids in the unbleached shells, followed by Ala, Ser and Thr (Supplementary
information Fig. S1a). Upon bleaching, Pro (proline) was the most abundant amino acid (Huang et al., 2023), but this
secondary amino acid is not quantified in the current analytical method used for AAG. As in our bleaching experiments, in
the work of Huang et al. (2023) upon bleaching Gly decreased in composition, while Asx, Glx, Ala, Val, Arg, Phe showed an
increase in composition; other amino acids present in lower concentrations show no or opposite trend. There is therefore a
general agreement between the study from Huang et al. (2023) and our current work, with the differences possibly due to the





sampling approach: Huang et al. (2023) homogenised all three aragonitic layers after removing the periostracum, whereas our
study separates the oOSL, iOSL and ISL, providing a more detailed study of the amino acid composition in the three
microstructural layers.

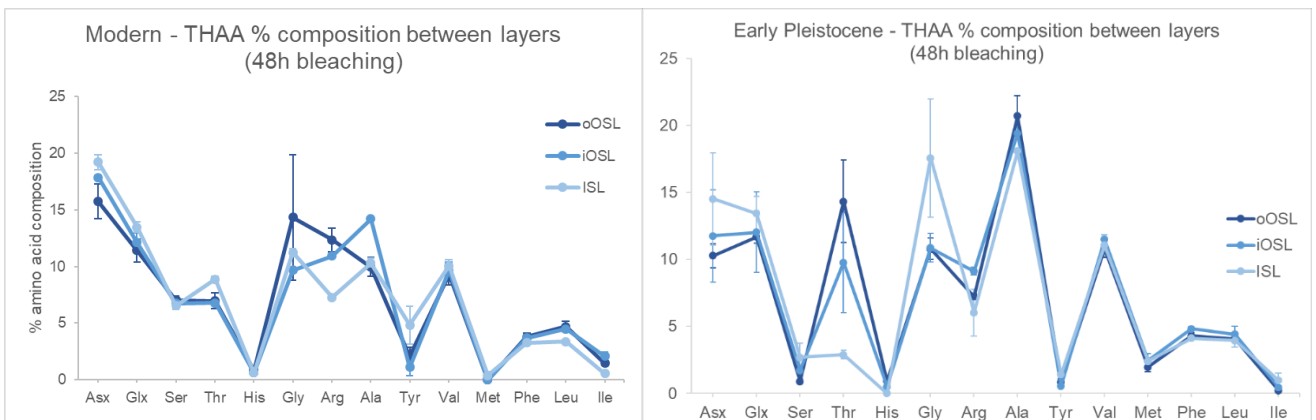

Figure 7. Mean THAA percentage composition in the three microstructural layers of *A. islandica* from (left) a modern sample
from Peterhead, and (right) an Early Pleistocene shell from Weybourne Crag. Error bars represent one standard deviation
based on three replicates for the shell from the modern shell, and two replicates from the Early Pleistocene shell. There are
differences in amino acid composition between the three aragonitic layers in the modern and Early Pleistocene shells,
indicating differences in original protein composition.
**3.3 Elevated temperature experiments to test for closed system behaviour**
High temperature experiments are considered a controlled, simple way to assess the suitability of biomineral proteins for AAG
(Kriausakul and Mitterer, 1978; Haugen and Sejrup, 1992; Penkman et al., 2008; Hendy et al., 2012: Demarchi et al., 2013).
The resistance of the IcP to oxidation was shown with bleaching experiments on modern and Pleistocene shells (Sec 3.2). To
test whether the IcP behaves like a closed system, the bleached powder (48 h) from the three layers of a modern shell of *A.
islandica* from the North Sea (ArPe, Table 1) was exposed to high temperatures in hydrous conditions (140°C for 8, 24, 48 h),
and the protein degradation (including rates of racemisation) observed. The high temperature experiments are utilised to
accelerate the protein degradation and explore the processes that would otherwise occur over thousands of years. Previous
studies showed that the degradation patterns in high temperature heating experiments do not necessarily produce the same
degradation patterns in subfossil samples; low temperature data (~80°C) may be more similar to subfossil results but requires
long exposure (Crisp et al., 2013; Tomiak et al., 2013; Demarchi et al., 2013). Nevertheless, the chosen temperature of 140°C
allows for quick assessment of protein degradation patterns and leaching over short timescale (a few days), while trends in



concentration and D/L values, and correlations of FAA and THAA D/L with increased exposure to 140°C can provide evidence
on whether the amino acids in *A. islandica* behave as a closed-system (Penkman et al., 2008).
The total concentration of FAA in the intra-crystalline fraction increases over time because prolonged heating breaks the
peptide bonds to ultimately release free amino acids (Fig. 8). The total THAA concentration decreases with heating due to the
decomposition of amino acids (Penkman et al., 2008; Crisp et al., 2013; Tomiak et al., 2013; Demarchi et al., 2013), discussed
in detail below. An interesting observation is that the ISL layer has a higher THAA concentration than the other layers, but
also has a steep increase in FAA concentration with heating time, coinciding with a steep drop in THAA concentration. This
means that the amino acids in the inner layer (ISL) are more susceptible to peptide bond hydrolysis, and in the hydrolysable
fraction (which includes both bound and free AAs), they are more prone to decomposition than the outer layer. This may be
due to differences in the proteins' primary sequence or higher structures, or a result of the way the proteins are mineral-bound
in the ISL microstructure. Conversely, the concentration of FAA and THAA in the iOSL layer shows the least change,
indicating that the protein in this layer may be more resistant to degradation.
The different rates of breakdown of the three layers indicate the importance of consistently sampling one microstructure for
obtaining more reliable AAG results.

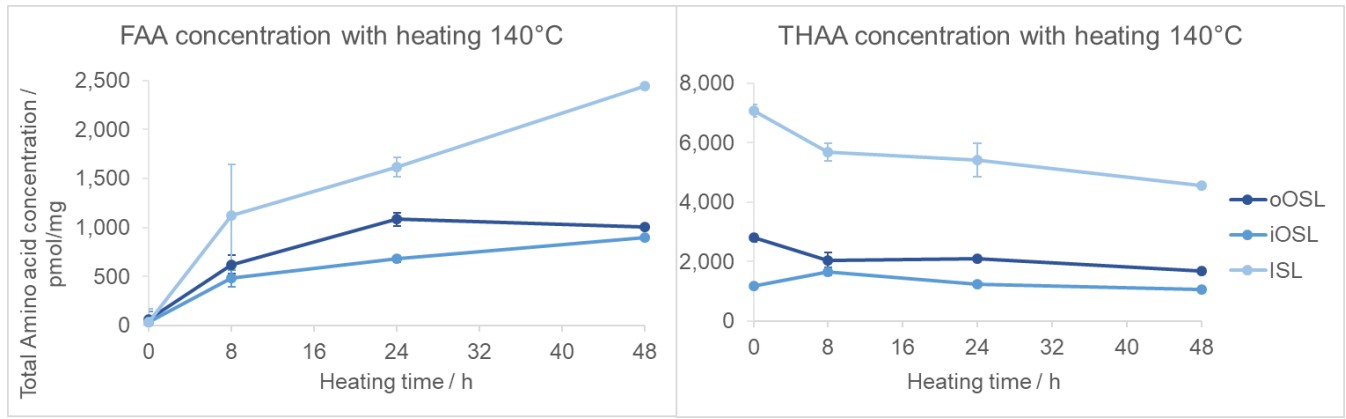


Figure 8. FAA and THAA concentration changes with heating at 140°C in the three shell layers of modern *A. islandica*. Error
bars indicate one standard deviation based on three replicates. The FAA concentration increases with heating due to peptide
bond hydrolysis; the ISL seems to have faster peptide bond hydrolysis compared to the other layers.
If *A. islandica* resembles a closed system the diagenetic products of protein degradation would be retained, and thus the FAA
and THAA D/L would be highly correlated (Preece and Penkman, 2005; Penkman et al., 2007; Demarchi et al., 2011, 2015.
As expected, the D/L values for all amino acids increase with increased heating duration in all layers (Fig. 9a, Supplementary



information Fig. S3) meaning that racemisation patterns follow reliable trends in the intra-crystalline protein fraction in *A. islandica*. Figure 9b shows the correlation of FAA and THAA for Asx, Glx, Ser, Ala, Tyr, Val and Phe: overall, all amino acids from all layers show high covariance indicative of closed-system behaviour. However, some scattering is observed for the ISL layer. There is also some scattering for Ser especially in the outer layers, which is expected in these high temperature experiments (Bright and Kaufman, 2011; Crisp et al., 2013) because the thermally unstable Ser (the "parent") readily degrades into Gly and Ala (the "products"). It is expected that the ratio of the "parent" over a degraded product will decrease with heating and thus indicate increased decomposition (Bada et al., 1978). This is particularly evident after 8 h heating with a marked reduction in the ratio of [Ser]/[Ala] (Fig. 10). In the THAA fraction Ser D/L decreases after 24 h (Supplementary information Fig. S4) due to decomposition of free serine, resulting in a decrease in the overall racemisation of Ser (Penkman, 2010).

Other decomposition pathways include the degradation of Ser, Thr and Tyr (the "parent") into Gly (Vallentyne, 1964), and Asx (the "parent") into Ala (Walton, 1998). These trends were observed in all cases in the FAA and THAA samples of the iOSL layer after 8 h and in the oOSL layer after 24 h, whereas in the ISL layer the ratios increased in some cases (Supplementary information Fig. S4). As previously mentioned for the concentration and D/L values, this could either be due to how the different peptides are bound to the mineral, or differences in protein sequence and structure in the ISL layer compared to the outer layers. Upon heating, the concentration of FAA increases at a high rate in the ISL (Fig. 8) and the steep "parent"-product ratios may reflect the more labile nature of the peptide bonds. The THAA composition of the bleached unheated ISL layer also shows a higher percentage of the more labile "parent" amino acids Asx, Thr and Tyr compared to the outer layers, while Ser has similar composition in all three layers (Supplementary information Fig. S5). Therefore, the high proportion of amino acids with labile peptide bonds in the ISL explains the high decomposition rate of FAA (Fig. 8) and the faster degradation rates.

The high correlation between FAA and THAA amino acid D/L values and the predictable degradation pattern observed from the high temperature experiments point towards a closed-system behaviour for *A. islandica* in all three layers However, these differences in rates of degradation between the inner and outer layers would affect the D/L values and the accuracy of the AAG interpretation, therefore it is preferable to analyse one specific layer. Interestingly, in the ISL a high proportion of amino acid is lost to hydrolysis in the THAA fraction and to degradation in the FAA fraction. In isotope studies the oOSL is not used because it can be more readily contaminated or impacted by environmental factors, it being the most external layer (Schöne, 2013). The iOSL is routinely used in isotope analyses and can be used in sclerochronology (Schöne and Huang, 2021; Butler et al., 2009). Due to the previous research on the iOSL, our bleaching and high temperature degradation experiments and ease of sampling the iOSL, we therefore suggest using this same layer for AAG.



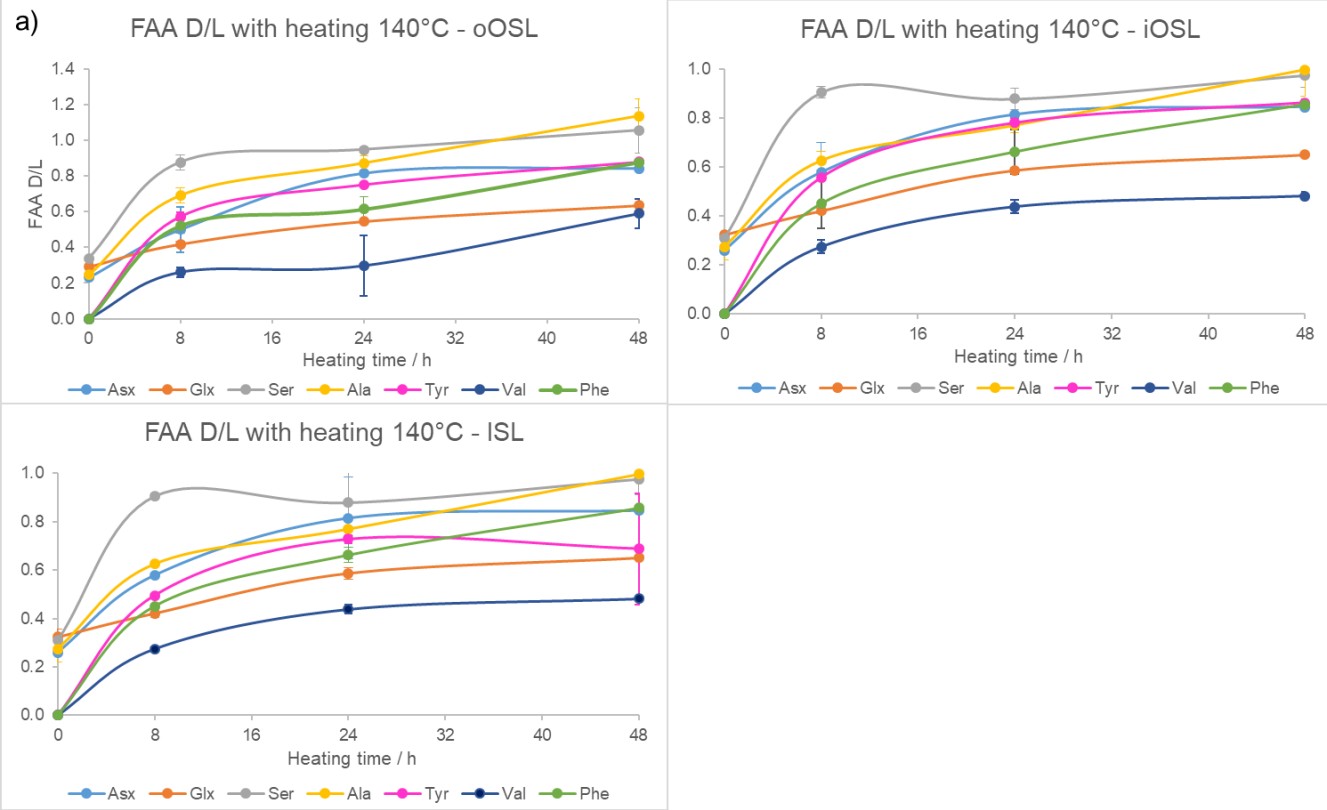

383





Figure 9. (a) Mean FAA D/L with increased duration of heating at 140°C in the three bleached layers of modern *A. islandica*; error bars indicate one standard deviation based on three replicates. (b) FAA vs. THAA D/L with heating at 140°C. D/L values increase with increasing exposure to high temperature in all three aragonitic layers, and high correlation between the FAA and THAA fractions in most cases.





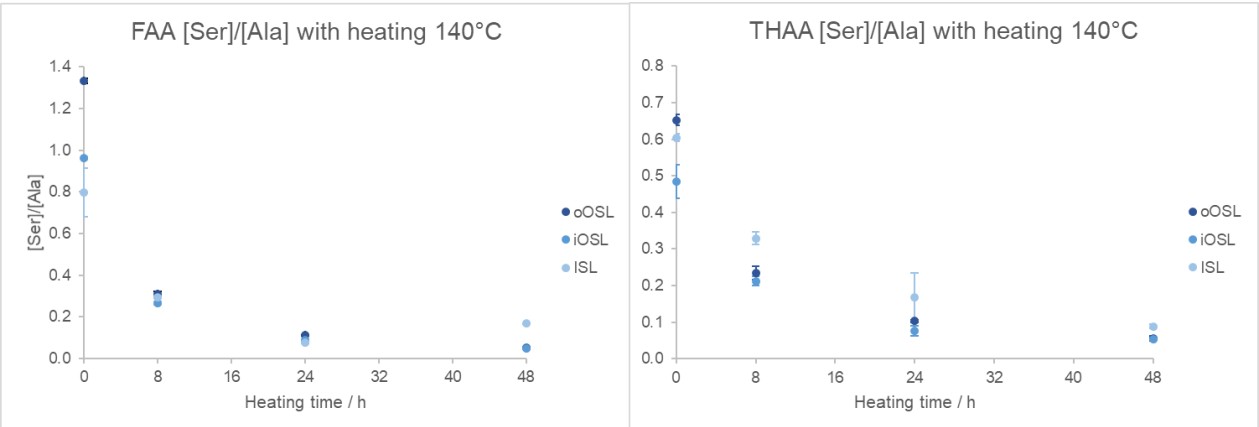

Figure 10. [Ser]/[Ala] in the bleached oOSL, iOSL and ISL layers of modern *A. islandica* from Peterhead following heating at 140°C for 8-48 h.  Error bars indicate one standard deviation based on three replicates.  The [Ser]/[Ala] decreases with heating in all three aragonitic layers.

**3.4 Assessing ontogenetic trends in modern and subfossil AAG**

Previous work on amino acid δ$^{15}$N of *A. islandica* has shown changes in isotope values and amino acid composition with ontogeny, i.e. with biological age of the shell (Schöne and Huang, 2021).  Here, eight shells with known ages spanning 100-400 years (Table 1) were sampled near the hinge (representing the early ontogenetic age of the shell) and near the margin (representing late ontogeny), to check for any differences in composition and D/L values.  Given the importance of original protein composition to the subsequent degradation, it is important to determine whether there are differences in concentration and D/L between early and late ontogeny as seen in amino acid isotopic analyses (Schöne and Huang, 2021).  In addition, if the rates of the reactions are fast enough, it may be possible to use AAG for age resolution *within* an individual shell.  For example AAG has been used in sclerochronological studies of tropical Porites corals (e.g. Goodfriend et al., 1992), providing a resolution of ± 6 years in most recent material and ± 24 years in the last 150 years (Hendy et al., 2012).  In those cases the ability to obtain high resolution data was due to the relatively high ambient temperatures (~26°C; Hendy et al., 2012) in which the corals live, but the lower temperatures of *A. islandica's* habitat (~1-16°C; Schöne, 2013) mean that AAG for sclerochronology may not be applicable to this biomineral.

It is expected that the iOSL of samples from early ontogeny will have higher D/L values, because this part of the biomineral would have been deposited earlier in time; late ontogenetic samples will have lower D/L values.  As the fastest racemising amino acids (Fig. 9a; Supplementary information Fig. S3), Asx, Ser and Ala were examined in detail (Fig. 11).  The error bars are quite large in the FAA samples, likely due to the low concentrations of amino acids, so the data should be treated with





caution and therefore only the THAA are going to be discussed here. In the Mid-Holocene samples the D/L values for Asx, Ser and Ala show higher values in late ontogeny, contrary to the expectation (Fig. 11a). The intra-shell variability is very low (σ=0.005-0.02 for FAA and THAA), and the lack of ontogenetic trend is likely related to the older age of the shells confounding the *in vivo* degradation. The post-medieval shells do not show any significant ontogenetic pattern (Fig. 11b). The expected higher D/L values in early ontogeny are present in the modern shells FAA Asx, THAA Asx and Ser D/L plots, but not for Ala (Fig. 11c). Similarly to our data, Goodfriend and Weidman (2001) showed a gradual decrease in D/L in the unbleached outer layer of modern *A. islandica* shells from the umbo to the rim, but the trend was less evident in subfossil shells, especially in increments older than $1050 \pm 35$ ($^{14}$C age). The increments in early ontogeny also showed a much higher extent of racemisation connected with fast growth and large band ages compared to the rest of the shell, indicating that there are different proteins responsible for shell growth in early and late ontogeny (Goodfriend and Weidman, 2001). As a result, they recommended consistent sampling of the iOSL layer in late ontogeny or at least after increment year 20 (Goodfriend and Weidman, 2001).

It is notable that the D/L values in the THAA fraction of modern samples in early ontogeny follow the year of birth in the THAA fraction (Fig. 11c): meaning that the eldest shell that settled from larva first in 1865 has highest D/L (Fig. 11b, M1, blue circle), followed by the shell represented by the orange circle (Fig. 11b, M2) settled in 1874 and the least racemised sample is the shell that was settled in 1908 (Fig. 11b, M3, grey circle). For the post-medieval shells, intra-shell variation is high, especially for the D/L values corresponding to ~ 1400 CE. Similarly, the late ontogeny modern shells (Fig. 11c, M1, M2, M3) all died in 2004 and should have similar D/L values, but they show great variability and/or large error bars, except for the FAA Asx D/L values.

The concentration of amino acids is higher in early ontogeny samples in the Md-Holocene shells from the North Sea, whereas the opposite trend is observed in the post-medieval and modern shells (Supplementary information Fig. S6). Goodfriend and Weidman (2001) observed a slight decrease in percentage composition of Ser, Tyr, Met, Ile and Leu with age and an increase in Glu, Val, Ala and Asp with age. Overall, there is no specific trend in composition with ontogeny in our shells, although some of the palaeontological and modern shells show similar results to Goodfriend and Weidman (2001), indicating that there may be more acidic intra-crystalline proteins responsible for growth during early ontogeny compared to late ontogeny, where basic amino acids are more prominent (Supplementary information, Fig. S6). The different proteins in early and late ontogeny may also be responsible for the variability in D/L. In summary, the D/L values in early and late ontogeny of modern, post-medieval and Mid-Holocene age have high intra-shell and inter-shell variability, suggesting that AAG is not suitable for providing *within*-shell chronologies in *A. islandica* shells. Given the possible variability in D/L values and protein composition with ontogeny, it is recommended to consistently sample the iOSL layer for AAG; late ontogeny is preferred because of the increased thickness of the iOSL layer.







Figure 11. THAA Asx, Ser and Ala D/L for (a) Mid-Holocene, (b) post-medieval, (c) modern *A. islandica* early and late ontogeny samples. Note: the age sampled for AAG may vary slightly from the sclerochronological age reported. Error bars indicate one standard deviation based on two analytical replicates. Except for the modern samples, AAG shows no ontogenetic trends

**3.5 Optimised method and recommendations**

The bleaching experiments have shown that the IcP of all three microstructural layers can be isolated after 48 h of bleaching (Figs. 4-5; Sec. 3.2). Heating experiments showed that all layers behave as a closed system. The ISL has a higher rate of peptide bond hydrolysis (Fig. 8), likely due to the higher percentage composition of labile amino acids compared to the outer layers. The slightly higher scattering in D/L values in the ISL (Fig. 9b) suggests the use of the outer shell layer for future



dating. The low peptide bond hydrolysis and co-variance between FAA and THAA in the oOSL and iOSL suggests that these
layers may provide more reliable dating (Sec. 3.3). Given that the iOSL is easier for sampling, as this layer is the widest, and
is already used in sclerochronological and isotope studies, we recommend using this layer for AAG. From the ontogenetic
trends observed (Sec. 3.4), it is recommended to sample the late ontogeny (near the margin) portion of the iOSL; this should
ensure more consistent protein analysis.
In conclusion, for AAG analysis of *A. islandica* we recommend cleaning of the shell in deionised water with sonication and
selective drilling of the iOSL from a portion deposited in late ontogeny. The drilling step can be done by slicing the shell from
the umbo to the margin (Sec. 2.2), and then either selectively drilling the iOSL with a hand-held rotary burr if this layer is
thick, or drilling the oOSL away until the iOSL is reached and collecting only the latter layer. Caution needs to be taken to
continuously move the rotary burr to reduce the build-up of temperature that can degrade the protein (Sec. 3.1). The powdered
iOSL is then exposed to NaOCl for 48h and removed by washing with water and MeOH. The demineralisation, hydrolysis
and UHPLC analysis steps are outlined in section 2.
**3.6 An initial IcPD AAG framework for *A. islandica***
Following the isolation of a stable intra-crystalline protein fraction that shows effectively closed-system behaviour in the iOSL
of *A. islandica* in laboratory experiments, we analysed subfossil shells with independent evidence of age to observe the amino
acids' degradation patterns in *A. islandica* during the Quaternary period. The Quaternary shell samples used in this initial
framework were *A. islandica* already dated by sclerochronology, radiocarbon dating, subfossil evidence and AAG of other
material in the same horizon (Table 1; Butler et al., 2009, 2013; Estrella-Martinez, 2019; Preece et al., 2020; Supplementary
information Table S2). The iOSL was sampled, when possible, from late ontogeny for consistency of results. Samples were
prepared as outlined in sections 2 and 3.5.
In a closed-system the FAA and THAA D/L values are highly correlated and indicate that both fractions of amino acids degrade
predictably. From the high temperature experiments (Sec. 3.3) Asx and Ser were the fastest racemisers, meaning that they
should provide higher temporal resolution for dating more recent specimens. Glx, Val and Phe show slower racemisation
(Supplementary information, Fig. S3), thus they may be able to date earlier in the Quaternary period. In our subfossil samples
the FAA and THAA D/L show a high co-variance for Asx, and good correlation of THAA Glx, Ser and Asx (Fig. 12),
indicating that subfossil samples follow a predictable degradation pattern. In some amino acid parameters there seems to be
different degradation patterns when comparing the high temperature experiments and subfossils (Supplementary information
Fig. S7). This has been seen before in other biominerals (e.g. Tomiak et al., 2013; Dickinson et al., 2019; Baldreki et al.,
2024), and may be due either to limitations of these high temperature experiments, or different degradation pathways which
are enabled under high temperature conditions. Both preclude using this high temperature dataset to calculate kinetic



parameters for this biomineral. While the IcP framework is richer in the Holocene period and very limited for the Pleistocene,
the Early Pleistocene and Mid-Holocene samples are well-separated for all amino acids presented here, showing that it is
possible to distinguish between Pleistocene and Holocene samples using *A. islandica* (Fig. 12). However, using Glx and Val
(Figs. 12a, c, d) it is not possible to distinguish the modern and post-medieval shells. In the Asx and Ser plot (Fig. 12b) the
modern, post-medieval and Late Holocene (Walker et al., 2019) shells are better separated, although some overlap is still
present. In this plot, the THAA Ser values for the Early Pleistocene shells are lower than in modern shells, because free Ser
naturally decomposes with age as previously shown in the heating experiments (Sec. 3.3) and in other biominerals (Penkman
et al., 2008; Penkman, 2010; Crisp et al., 2013; Demarchi et al., 2013, 2015).







Figure 12. a) FAA vs. THAA Asx D/L; b) THAA Asx vs. Ser D/L; c) THAA Asx vs. Glx D/L; d) THAA Asx vs. Val D/L; e) THAA Asx D/L vs. age and inset focusing on the last 8 ka; f) THAA Ser D/L vs. age of modern, post-medieval, Late Holocene, Middle Holocene and Early Pleistocene *A. islandica* shells. D/L values for the slower racemising amino acids (e.g. Glx) span the Quaternary period, while the faster racemising amino acids (e.g. Asx, Ser) allow temporal resolution within the Holocene.





These preliminary results indicate that it is possible to use the IcP in the iOSL of *A. islandica* for AAG of Quaternary shells.
The Early Pleistocene shells have very high D/Ls for the fast racemiser Asx in (FAA Asx D/L ~0.85; THAA Asx D/L ~0.69),
approaching the end-point for using Asx in AAG (Torres et al., 2013; Demarchi et al., 2013). Glx and Val D/L values were
lower (FAA Glx D/L ~0.63, Val D/L ~0.75; THAA Glx D/L ~0.50, Val D/L ~0.56) meaning that there is potential to use these
slower racemisers to date shells further back into the Pleistocene and Late Pliocene (Fig. 12c, d; Penkman et al., 2007; Reichert
et al., 2011; Hendy et al., 2012; Torres et al., 2013; Demarchi et al., 2013; Millman et al., 2022). On the Holocene timescale,
the fast racemisers Asx and Ser provide reliable D/L separation between the Middle and Late Holocene (Fig. 12f). Modern
samples have a slight overlap with post-medieval shells inTHAA Ser and Asx, meaning that the resolution of AAG for *A.*
*islandica* for these amino acids may be approx. 1500-2000 years during the Middle and Late Holocene in the temperate-cold
climate of the North Sea. Given the non-linear nature of AAG, the resolution will be reduced into the Pleistocene, but further
analyses are required to assess the resolution. If samples date from the last ~50 ka, then radiocarbon dating will provide a
higher resolution dating method compared to AAG in the temperate-cold environment where *A. islandica* typically lives,
although it requires correction for the marine reservoir effect (Hajdas et al., 2021). Nevertheless, AAG using the IcP of the
iOSL of *A. islandica* has the potential to discriminate Middle and Late Holocene samples, and further back into the Early
Pleistocene or Late Pliocene.
**3.7 AAG rangefinding of undated shells**
Since the Middle and Late Holocene, important cultural transitions and palaeoenvironmental and ecological changes, both
natural and human-induced, have taken place in the North Sea and Iceland and had an impact on the marine ecosystem (for
example the Mesolithic-Neolithic transition, the settlements of Vikings in Iceland, and the Industrial Revolution in Northern
Europe (Andersen, 2000; Ahronson, 2012; Poulsen, 2008). The palaeoenvironmental record contained within subfossil *A.*
*islandica* provides a unique way to study these important transitions, but dating is required to identify potentially relevant
shells. As part of the ERC SEACHANGE project, over twenty thousand *A. islandica* shells were collected from the North Sea
and Iceland seafloors during research cruise DY150 in 2022, with the aim to use these for geochemical and sclerochronological
studies (Scourse et al., 2022). Here we explore the potential for rangefinding age estimates of individual dead shells by AAG.
The initial IcPD AAG framework showed the potential to provide dating of shells with resolution of 1500-2000 years during
the Middle and Late Holocene. The rangefinding is expected to narrow down the age of the shells collected from the North
Sea and Iceland seafloors (Supplementary information Table S2).
The AAG age range finding was carried out on 160 shells (Fig. 13; Supplementary information Table S2). The AAG dating
determined that these shells likely span the Middle and Late Holocene, with both the Asx and Ser D/L values in agreement
with this time period. In cases where there was agreement between the three most useful parameters for the Holocene (FAA



Asx D/L, THAA Asx D/L, and THAA Ser D/L), the narrowest age range possible was assigned (Fig. 13). It is noted that there
are a few shells that overlap between age periods, likely due to the resolution of AAG. In case of agreement of two of the
three D/L values, a wider age range was assigned. For example, shell Ic22200193 showed correlation with Late Holocene for
the THAA Asx and Ser D/L, but the FAA Asx D/L value overlapped between the Late Holocene and post-medieval age; due
to the agreement of two of the three parameters with the Late Holocene (which includes post-medieval), this shell was assigned
an age range correlating with this stage. Shell FG22202523 showed THAA Asx D/L indicating a modern age, but FAA Asx
D/L and THAA Ser D/L overlap between modern and post-medieval age, thus a post-medieval-modern age range was assigned.
These three screening methods resulted in 93 shells with a narrower age range and 67 shells with a wider age range (either
because of agreement of two of the three D/L, or overlap of D/L values between age ranges). There were four shells,
Ic22201300, Ic2220035, Ic22200194, Ic22202048, which showed D/L values consistently higher than the Late Holocene shells
but lower than the Mid-Holocene shells; thus, they were categorised as older than ~4 ka and younger than ~8 ka in age (8
ka<shells>4 ka). Ten shells from the Fladen Ground exhibited THAA Asx and Ser D/L values slightly higher than the Mid-
Holocene D/L values, thus they were categorised as Early Holocene or older. The results of AAG rangefinding of *A. islandica*
shows that this technique is able to narrow down the ages of shells, assigning 10 shells to the Early Holocene or older, seven
shells as younger than ~8 ka and older than ~4 ka (8 ka< shells>4 ka), 23 shells to the Late Holocene, 34 shells to post-medieval
age, and 86 modern shells. These analyses provide an initial age range for *A. islandica* shells that, depending on the time
period of interest, can then enable selection of appropriate shells for more accurate dating with radiocarbon,
sclerochronological crossdating and studied for palaeoecological information.

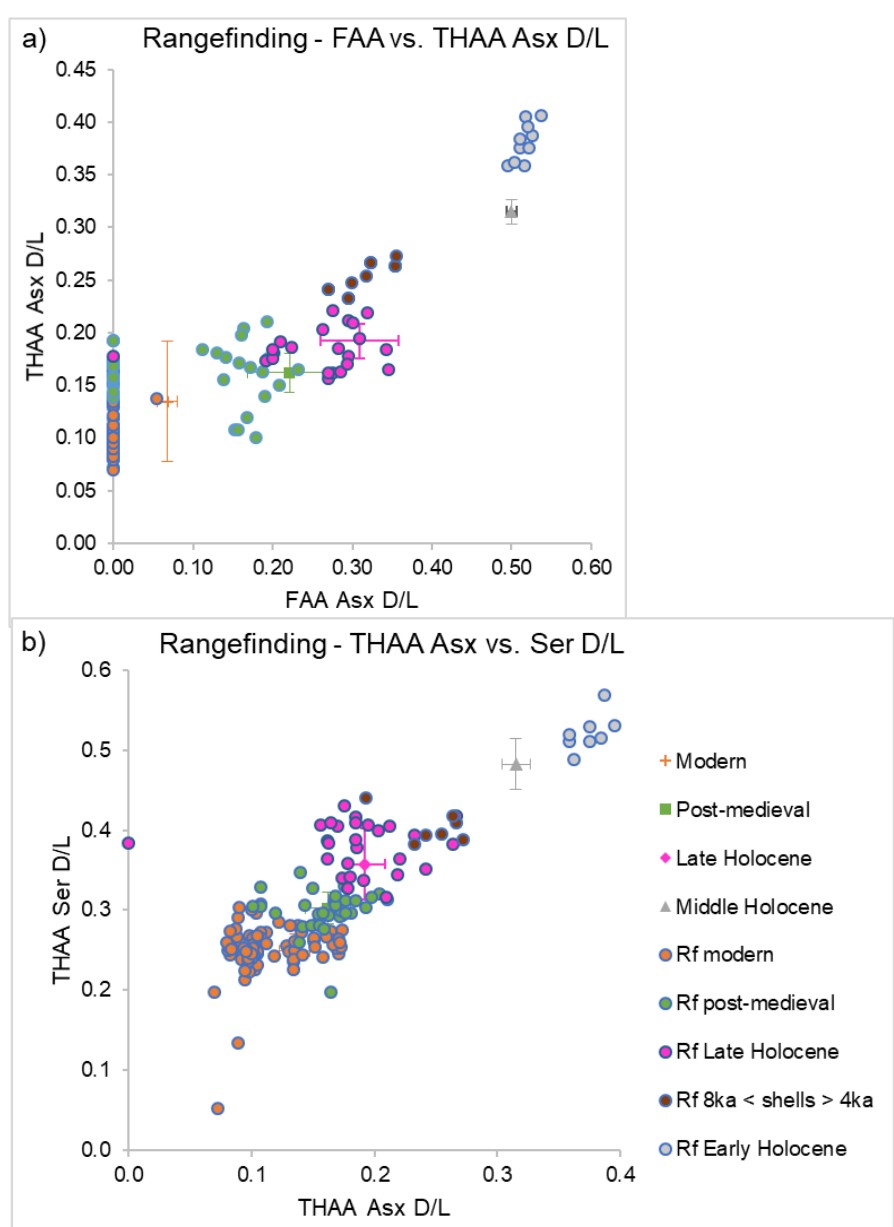

Figure 13. Rangefinding (Rf) of *A. islandica* shells within the IcPD framework (a) FAA vs. THAA Asx D/L, (b) THAA Asx vs. Ser D/L. Modern samples are in orange, post-medieval in green, Late Holocene shells in pink, shells older than ~4 ka and younger than ~8 ka (8 ka < shells > 4 ka) in brown, and Early Holocene in grey.



**4 Conclusion**

A protocol for the analysis of intra-crystalline chiral amino acids for amino acid geochronology (AAG) of the bivalve *A. islandica* has been established. The three-layer microstructure of the shell has been investigated to determine which layer would be most applicable to AAG. The intra-crystalline protein (IcP) fraction was successfully isolated with NaOCl oxidation for 48 h. This analysis highlighted different amino acid compositions between the three layers (oOSL, iOSL and ISL), meaning that for reliable dating a single microstructural layer should be sampled. Heating experiments at 140°C showed that the protein fraction in the inner layer ISL is more prone to peptide bond hydrolysis than the outer layers, possible due to the high composition of labile amino acids in this layer. Conversely, the outer layers show low loss and decomposition of amino acids. Nevertheless, all three layers show good co-variance between FAA and THAA D/L and behave as a closed system. The iOSL layer is recommended for AAG because it is already used for isotopic and sclerochronological studies. The oOSL, the outermost layer, is more exposed to the external environment and marine organisms and is thinner than the iOSL, thus harder to select. Samples of early and late ontogeny in modern, post-medieval and Mid-Holocene shells did not show a consistent pattern of composition and D/L, thus the resolution and sensitivity of AAG is too low for sclerochronological studies *within A. islandica* shells of this age. The optimised method of analysis of the iOSL, following bleaching for 48 h, was applied to Quaternary subfossils, providing an initial dating framework, with the fast racemisers Asx and Ser able to distinguish Mid-Holocene from post-medieval/modern samples, providing a tentative resolution for AAG of *A. islandica* of approx. 1500 years in the Late Holocene. The slower racemising amino acids are able to date back to at least the Early Pleistocene. Rangefinding of 160 undated shells showed that AAG can securely separate between modern, post-medieval (~1100-1700 CE), Late Holocene (4-1 ka), Mid-Holocene (~8 ka) and Early Holocene (>8 ka) shells. Further analyses are required to expand the framework and better establish the age resolution for this biomineral, but these initial promising results indicate that *A. islandica* is a reliable biomineral for AAG dating of marine deposits during the Quaternary period and for rangefinding collections of *A. islandica* shells of unknown age.

**Author contribution**

Conceptualisation: Kirsty E. H. Penkman, James D. Scourse

Data curation: Martina L. G. Conti

Formal analysis: Martina L. G. Conti, Kirsty E. H. Penkman

Funding acquisition: Kirsty E. H. Penkman, James D. Scourse

Investigation: Martina L. G. Conti, Kirsty E. H. Penkman

Methodology: Martina L. G. Conti, Kirsty E. H. Penkman



Project administration & supervision: Kirsty E.H. Penkman, James D. Scourse
Resources: Kirsty E. H. Penkman, Martina L. G. Conti, Paul G. Butler, David J. Reynolds, Tamara Trofimova, James D.
Scourse
Validation: Martina L. G. Conti, Paul G. Butler, David J. Reynolds, Tamara Trofimova, James D. Scourse, Kirsty E. H.
Penkman
Visualisation: Martina L. G. Conti, Kirsty E. H. Penkman
Writing - original draft preparation: Martina L. G. Conti, Kirsty E. H. Penkman
Writing - review and editing: Martina L. G. Conti, Kirsty E. H. Penkman, Paul G. Butler, David J. Reynolds, Tamara
Trofimova, James D. Scourse
**Competing interests**
Kirsty E. H. Penkman is an associate editor of the journal.
**Acknowledgments**
The authors declare that they have no conflict of interest.
The SEACHANGE Synergy Project has received funding from the European Research Council (ERC) under the European
Union's Horizon 2020 research and innovation programme (Grant Agreement No 856488).
Iceland radiocarbon dates were funded by the EU Framework 6 MILLENNIUM Integrated Project 'European climate of the
last millennium' (SUSTDEV-2004-3.1.4.1, 017008-2).
North Sea radiocarbon dates were funded by European Union Fifth Framework HOLSMEER project (EVK2-CT-2000-00060)
and the United Kingdom Natural Environment Research Council standard research grant (NER/A/S/2002/00809)
Thanks to Mr. J. Scolding, Dr. Anna Genelt-Yanovskaya and Dr. R. Preece for providing some of the samples. Thanks to Dr.
Niklas Hausmann for discussing initial sampling techniques and for providing samples. Dr S. Presslee and Mr. M. von Tersch
are thanked for initial laboratory training and Ms. S. Taylor for administrative support. Many thanks to Dr. Lucy Wheeler, Dr.
Marc Dickinson & Ms. C. Bäldreki for helpful comments on an initial version of this manuscript.



**Open access policy**

For the purpose of open access, the author has applied a Creative Commons Attribution (CC BY) licence to any Author Accepted Manuscript version arising from this submission.

**Data availability**

Data in this study has been included in the Supplementary information Table S2 and all amino acid data from this study will be made available through the NOAA repository upon publication: ftp://ftp.ncdc.noaa.gov/pub/data/paleo/aar/.

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
