# Peer review of "A new method for amino acid geochronology of the shell of the bivalve"

_EGUsphere, 2023_

## Author Comment (AC1)

**Responses to anonymous referee 2**

We thank the reviewer for their useful feedback and valuable comments that will improve the manuscript.  Please find below our responses.

The reviewer's comments are in black, our responses in *blue italics* and any suggested changes/additions are in *red italics*.

1.   A new and higher quality location map should be included. The sample locations should be identified with a number on the map.
*A new map created with ArcGIS was added to Figure 2 and sample locations are identified with letters "A" to "U" and the rangefinding samples as "Rf".*

[Figure]

*Figure 2. a) Location of the A. islandica samples analysed in this work.  Map created using ArcGIS Pro.*

2.   Each sample should be identified by numbers/sample siets in the table.
*All samples are now identified with letters "A" to "U" and the rangefinding samples as "Rf" in Table 1 and Figure 2a.*

3.   A scetch of a cross section illustrating the growth of time with should be included. This will illustrate the way the inner layer(which is growing through the life time is making up most of the section  when approaching the umbo).
*Thank you for this helpful suggestion, we have added a simplified schematic of the growth of the OSL, ISL and hinge in figure 1c.*

[Figure]

Figure 1. a) cross section of Arctica islandica showing the b) inner shell layer (ISL), inner portion of the outer shell layer (iOSL), and outer portion of the outer shell layer (oOSL); c) simplified schematics of the growth of ISL and OSL and hinge ("y" indicates the year of growth).

4.    Some places I do not follow the authors definition on when a plateau is reached in their experiments. For some of the data it seem like the plateau is reached between the first (initial) and second analytical points, and this should be  stated in text.
*We have clarified what we mean by plateau, defined as "little or no change in trend between two or more observations" in section 3.2.*

5.    It would be good if the authors somewhere present  some of their finding on a geological time line.
*We refer to the age of the shells to geological time periods (e.g. Pliocene, Pleistocene, Holocene) and the age of the shells is reported in Table 1.  We have added a timeline on a million-year scale for the Pliocene, Pleistocene and Holocene (Figure 2b), and thousand-year scale to the Holocene (Figure 2c).*

[Figure]

Figure 2. a) Location of the A. islandica samples analysed in this work.  Map created using ArcGIS Pro.  Geological timeline of the b) Pliocene and Pleistocene and Holocene on a million-year scale, and c) Holocene on a thousand-year scale (IUGS International Chronostratigraphic Chart, 2023).

6.   Also be clear if recent shells are found with a living animal inside.
*The only modern shell that was found with a living animal inside was the ArPe and ArPe2 specimens which were live-collected, meaning that the living animal was inside when the samples were collected.  However, we purchased the shells without the animal years after they were trawled.  Table 1 states that these samples were live-collected for other purposes, and it states the website we bought them from for this study.*

---

## Author Comment (AC2)

**Response to referee 1**
The reviewer's comments are in black, our responses in *blue italics* and any suggested changes/additions are in *red italics*.

*We thank reviewer 1 for taking the time to provide valuable comments from a knowledgeable expert in the field that have improved our manuscript.  Please find below our comments.*

1) In my view the authors should amplify the discussion especially Sections 3.2 and 3.3 regarding the use of more samples.
In section 3.2 the authors used a modern sample and an early Pleistocene sample to test the impact of bleaching. In my view the authors should also use:

- Not just a sample of each of them (modern and early Pleistocene), but more shells to be statistically reliable.

- They should also use medieval and Holocene samples.

*We have carefully considered the reviewer's request for additional experimental samples, suggested for the bleaching study.  We understand the reviewer's request for replication to make sure that the results we are obtaining are not just specific to a single sample. However, as we had to analyse 3 layers from each shell (rather than the usual single layer), this tripled the complexity of the usual experimental undertaking.  We feel that we have captured the individual uncertainty through the use of replication on samples of different ages.  Our arguments are presented below:*

*The logistics of these experiments were magnified by the need for 3 individual layers to be analysed per experiment, rather than for most other bleaching experiments where only a single layer was analysed (e.g. Penkman et al., 2008; Hendy et al., 2012; Demarchi et al., 2013a-b; Bridgland et al., 2013; Ortiz et al., 2013; 2015; ; Tomiak et al., 2013; 2016; Crisp et al., 2013; Dickinson et al., 2019; Baleka et al., 2021; Wheeler et al., 2021). This tripled the number of analyses that were required, tripling preparative time and analytical time.  Each of these layers was also done in triplicate.  Given the high number of analyses that needed to be undertaken on a single shell, we therefore decided not to do experimental replicates using multiple shells of the same age, but instead to encompass a greater degree of variability, we undertook replicate experiments on a shell of a different age.   However in the revised version we have also added the data from additional bleaching experiments carried out on another shell that was beach-collected (and dated as modern in the framework, Table 1, Fig 4a), the data from which support our previous results.*

*With the presented data, we showed that the inter-crystalline fraction was present and that the intra-crystalline fraction was stable in modern and >2 Ma shells. Given the time and funding constraints, and the similarity in the results from shells that were not related in any way, we are confident that the patterns we observe are not just due to the behaviour of a single shell.  We therefore have not undertaken the requested additional replicate experiments, but we provide full experimental details so that others can*

*replicate the experiment. Since the submission of the original manuscript, we have also added to the Pleistocene dataset in the initial framework (Samples S, T & U, Table 1), which confirms our previous experimental findings, and have included these additional samples within this revision.*

2) In section 3.3 The authors performed the experiment using samples from the three layers of the same modern shell. In my view they should use more than one shell.

I also suggest extending the experiment beyond 48 hours.

*We understand the reviewer's request for replication to make sure that the results we are obtaining are not just specific to a single sample. However, as with the bleaching experiments, the logistics of these experiments were magnified by the need for 3 individual layers to be analysed per experiment.*

*We carried out heating experiments to check whether the intra-crystalline protein fraction generally behaves like a closed system and to study protein degradation under these extreme conditions. However we were focused on the early stages of diagenesis (rather than characterising the complete protein breakdown) as this was most relevant to our rangefinding study. As by 48h the fast-racemising amino acids, Asx and Ser, that were employed in the rangefinding of Holocene shells (section 3.7), had reached a plateau near equilibrium in the experiments, we decided not to extend the experiments further. The Asx and Ser D/L values from the rangefinding samples, as well as the D/L values reported in the framework, lie within the range covered by the high temperature experiments, and so cover the levels of degradation important for this work.*

*Given the time and funding constraints, and the similarity in the results from shells of the same age from the rangefinding, we are confident that the patterns we observe are not just due to the behaviour of a single shell. We therefore have not undertaken the requested additional heating experiments, but we provide full experimental details so that others can replicate the experiment.*

3) I recommend redrawing Figure 11 based on the comments provided below.

*Thank you for the suggestion, we have redrawn Figure 11 with the shell denomination (shells "A" to "U") from table 1.*

Other suggestions:

4) In some parts of the text, the authors use Asx and Glx, while in others Asp and Glu.

*The newly-collected data from our study only refers to Asx and Glx. We refer to Asp and Glu when the literature describes the data as Asp and Glu, i.e. Goodfriend and Weidman, 2001, and we now clarify this saying "aspartic acid as referred to by Goodfriend and Weidman, 2001".*

Section 3.1 Mineral diagenesis

5) I suggest changing the title of this section, since one of the most important conclusions is about the procedure for sampling the shells, which is not a diagenetic process.

*Thank you for the suggestion, we have changed the title to* "Sampling procedure and mineral diagenesis".

6) Line 190: The authors state that aragonite is the polymorph of CaCO3 that forms the shells of A. islandica. Is it valid for all three layers?

*Yes, all three layers are aragonitic.  Now clarified at the beginning of section 3.1, specifying* "Aragonite, the polymorph of $CaCO_3$ that makes up the three layers of the shells of A. islandica, can convert into calcite over geological timescales or under stress".

7) Lines 196: medieval or post-medieval?

*Thank you, we amended to* "post-medieval"

8) Lines 198-199. Also in post-medieval and mid-Holocene shells there is a small calcite peak.

*Yes, we now clarify this saying* "the Early Pleistocene shell (shell R, 2.2-2.1 Ma) shows only a very small calcite peak, compared to larger peaks in the modern and post-medieval shells".

9) Line 203. Regarding the process to remove the periostracum and isolate and sample the required layers, it is also possible to use HCl and not just drill.

*Thank you, we added the following sentence:*  "Other methods to remove the periostracum include using a scalpel, dipping the shell in HCl, NaOH or $H_2O_2$ (Checa, 2000; Agbaje et al., 2018), but it is challenging to isolate the individual mineralised layers without drilling."

10) Line 210. So, the presence of calcite in specimens up to Early Pleistocene is only due to the drilling procedure?

*Yes, that is our hypothesis as there wasn't any evidence of a calcite peak in the non-drilled sample. However differences in burial conditions mean that this may be sample-specific, so we have added this to the text:* "Individual burial conditions will impact the potential for mineral diagenesis in a sample, but it is also possible that the abrasion and

*temperature created during the drilling process may affect the aragonitic crystal structure".*

Section 3.2 Impact of bleaching

11) The authors used a modern sample and an Early Pleistocene sample to test the impact of bleaching. In my view, authors should use both:

- Not only one sample of each of them (modern and early Pleistocene), but more shells to be statistically realiable.

- They should also use medieval and Holocene samples.

*Please see the response above to comment 1.*

12) Line 215: There is and initial sharp decrease.

*Added "sharp"*

13) Line 218-219: with only one sample?

*Please see the response above to comment 1.*

14) Line 227: with only one sample?

*Please see the response above to comment 1.  The argument is strengthened with the evidence from the Early Pleistocene shell (sample R).*

15) Line 289: Which of the three layers should be used?

*At this point we can't suggest which layer to choose because the closed-system behaviour has not been studied (see section 3.3)*

Section 3.3 Closed system behaviour

16) The authors conducted the experiment using samples from all three layers of the same modern shell. In my opinion, they should use more than one shell.

*Please see the response above to comment 2.*

17) Line 325: the samples were exposed to high temperatures in hydrous conditions…under N2 atmosphere?

*This was not done under a nitrogen atmosphere because we are trying to replicate environmental degradation.*

18) Line 325: The authors heated the samples at 140ºC for 8, 24, 48h. Why not more times, up to 1 week or 10 days?

*Please see the response above to comment 2.*

Section 3.4 Ontogenetic trends in modern and subfossil AAG

19) The authors sampled eight shells from modern, post-medieval and Holocene times. They must indicate whether the samples come from the same layer (and which of the three layers was used) or come from the entire shell by mixing the three layers.

*We now clarify that only the iOSL was sampled, because it is the easiest to sample, and is the layer of choice for sclerochronology and isotope analysis.*

20) Were the samples taken from the same part of the shell? apex? Margin?

*We state that the iOSL was sampled near the hinge and the rim. The sentence says:* "the iOSL of eight shells with a known lifetime spanning 100-400 years (Table 1) were sampled near the hinge (representing the early ontogenetic age of the shell) and near the margin (representing late ontogeny), to check for any differences in composition and D/L values".

21) Lines 395-396. The authors indicate that the eight samples with ages spanning between 100 yr and 400 yr, but 3 shells were modern, 2 shells were from post-medieval times and 3 from the Holocene.

*Thank you - we have now clarified that 100-400 years is lifetime of shell (in vivo age) not date of shell (e.g. time since deposition).*

22) Line 401: cursive for Porites.

*Thank you, amended.*

23) I suggest redrawing Figure 11 and the caption should be explained better.

Various patterns were used to represent shell values from the Holocene and medieval and modern samples. They should be the same. In fact, they showed the D/L values for Holocene samples (delete them), but not for the other shells.

In the medieval sample plots the values were identified as ic 1 and ic2, but in the modern and Holocene sample plots this is not shown.

They should explain and unify the nomenclature of P1, P2, P3; M1, M2, M3; and ic1, ic2.

*Thank you, we redrew Figure 11 adding the shell letters reported in Table 1. The Mid-Holocene plot now matches the style of the other two plots. We also clarified what the figure shows in the caption.*

[Figure]

*Figure 11. THAA Asx, Ser and Ala D/L for (a) Mid-Holocene (shells O, P, Q), (b) post-medieval (shells G, H), (c) modern A. islandica (shells C, D, E) early and late ontogeny samples.  Note: the age of Mid-Holocene samples was assigned with radiocarbon dating, while for post-medieval and modern shells the age is based on sclerochronological cross-dating.  The age sampled for AAG may vary slightly from the sclerochronological age reported.  Error bars indicate one standard deviation based on two analytical replicates.  Except for the modern samples, AAG shows no ontogenetic trends.*

Section 3.6 IcPD AAG framewotk for A. islandica

 24) Line 470: were the samples taken from the apex?

*We stated that "The iOSL was sampled, when possible, from late ontogeny for consistency of results".*

Section 3.7 AAG rangefinding of undated shells

25) The authors should explain whether the analysis comes from the same layer or from the entire shell, and whether from the same part of the shell or form different parts.

*Clarified that* "The AAG age range finding was carried out on the iOSL laid down during late ontogeny".

26) Of course, that AAG is based on D/L values, but I suggest that you also consider the evolution of the concentration and mainly the percentages of amino acids in the discussion.

*Although in the manuscript we present just the D/L values for clarity, we did inspect all of the data (e.g. AA concentration, composition) as we agree with the reviewer that the whole protein decomposition is important.  Some of these graphs have now been added in the Supplementary information Table S2, and we have included the raw data in the Supplementary information Table S2.*

27) For more recent samples (modern and post-medieval), they must consider the "age at death" of the shell, as they can live more than 500 years.

*Thank you, this is an important point - all shells were sampled near the rim thus as close as possible to the age at death of the organism, in order to minimise any* in vivo

*racemisation. We now clearly state that* "The AAG age range finding was carried out on the iOSL laid down during late ontogeny."